# Is Scripture a Gift? Reflections on the Divine-Ecclesial Provision of the Canon

## Brad East 

College of Biblical Studies, Abilene Christian University, Abilene, TX 79699, USA; brad.east@acu.edu

**Abstract:** This article investigates whether the canon of Christian Holy Scripture is properly understood as a gift and, if so, what theological implications this might entail. Following the introduction, the article has three main sections. The first section proposes an expanded grammar by which to describe the production and reception of the canon in and by the church, under the superintending sovereignty of the divine will and action. The second section offers a guide to recent inquiry into "the gift" in the fields of philosophy and theology, particularly those theories that might prove useful for applying the concept of "gift" to Scripture. The third section unfolds a normative account of the Christian canon as a gift of the triune God to his people and through his people, thus making sense of the long-standing liturgical practice of responding to the reading of the sacred page in the public assembly with a cry of thanks to God.

**Keywords:** Bible; doctrine of Scripture; gift; gratitude; tradition; inspiration; church; liturgy; canon; theology

## 1. Introduction

"Thanks be to God!" Such is the church's response to hearing the Bible read aloud in worship. On its face the rationale for the cry of thanks is plain. For the lector does not merely read from the biblical text; she concludes the reading with a gloss, which hides in its formulaic simplicity a complex and indeed astonishing set of claims. Though formulaic, in other words, it is far from pro forma. The gloss is: "The word of the Lord". This phrase is a kind of liturgical suffix. It transforms in retrospect what the audience just heard. It is an ambassadorial announcement that those gathered in the assembly—whether thousands in a megachurch or hundreds in a cathedral or dozens in a storefront or two or three in a living room—have just been spoken to by the living God. How? Through the words of his servants: the written words of prophets and apostles, read aloud by pastors and priests of his people. These words are a medium, therefore, an instrument of the Lord's speech in the present tense and, hence, a means of grace. Just so, they are a mystery of the gospel: through them Christ is present to his beloved; through them Christ imparts blessings to the baptized; through them Christ fulfills his promise to his disciples: "I will not leave you orphaned; I will come to you" (John 14:18).[1] Christ comes to believers through the word: for, as the passages continues, those who love Christ keep his word; he abides in them, together with his Father (v. 23), since the word believers hear and keep is not Christ's own, "but is from the Father who sent me" (v. 24). The Father sends also another, "the Advocate, the Holy Spirit", whose work it is to teach believers and to bring to remembrance all that Christ taught while still on earth (v. 26). So that, in the hearing of the word in the assembly of faith, the Spirit is as it were the agent of advent; it is he who brings Christ to his people through his word, he whose presence effects union of the body with the head, he whose secret operation makes ordinary elements of creation and culture—water, bread, wine, texts—the visible and audible words of God. Which is to say, channels of divine charity.

No wonder, then, the reiterated liturgical response of thanks. What the church encounters each time Holy Scripture is read aloud in the liturgy is nothing less than the persons

and works of the Holy Trinity. What the church encounters, in a word, is *grace*. Grace is a gift, and a gift given elicits gratitude. In this respect liturgical gratitude expressed in reply to the reading of the word in Christian worship is a matter of common sense. What else should we expect?

I want to suggest that we take a second look, however. For it is not only divine grace or divine presence—in brief, the gospel itself—for which the church gives thanks in response to Scripture's reading. It is thanks for the reading as such. Which means that it is thanks for Scripture itself. Which raises a pregnant theological question: Is Scripture a gift?

That is the question I aim to explore in this article. For, on one hand, "the gift" is a much-debated concept in philosophy, ethics, and theology; as is, on the other hand, the nature or ontology of the canon. My goal, though, is less probative than it is probing. I aim not to demonstrate a claim so much as investigate a range of conceptual possibilities. Is it fitting to call the Christian Bible a gift? If so, in what sense is it a gift? What are the plausible ways of construing it as a gift? How does its gift-like character inform the church's use of and relationship to it? And what are the implications either for describing the church's long history with it or for prescribing proper (and proscribing improper) practices of receiving it?—precisely as a gift, that is, with the requisite gratitude, whatever that might entail.

I will proceed in three sections. First, I will offer a theological account of Scripture's artefaction. For the canon stands at the end of a long production line, and the agents of its manufacture are simultaneously divine and ecclesial. Here I will provide a sort of lexicon for this divine-ecclesial process, drawing on well-worn concepts (inspiration, illumination) while expanding the roster of terms for the larger set of moments and discrete activities that call for description. Second, I will offer a guide to the spectrum of theories on the gift and gratitude. This will not be exhaustive but only a reasonable sketch of the main positions on offer so that we are able to see what it would mean, from different vantage points, to understand the Bible as a gift. I want less to commit to a single master theory of Scripture-as-gift than to hold open multiple conceptual paths: partly because I am less confident than others of the one true gift-theory; partly because this investigation is exploratory, and therefore patterned on the form of plural "if . . . then" queries rather than a single "because . . . therefore" argument. Third and finally, having provided both a set of ways of speaking of the canon as an artifact of divine-ecclesial provenance and a map of ways of speaking of the gift in general, I will suggest some conclusions about thinking and speaking of Holy Scripture as a gift of God for the people of God. My hope is to edge some small degree closer to understanding why it is, and why it is so fitting, that the people respond as they do: "Thanks be to God!"

## 2. The Making of Scripture

The doctrine of Scripture, or bibliology, contains many elements. The origins or making of Scripture is only one of them. Others include the nature of Scripture, its attributes or properties, its purpose in the mission of the church, and its proper interpretation. About these and related matters I will say little in what follows. I am interested exclusively in what I will call the artefaction of the canon.

For the canon, as is often said, did not drop from heaven. It is a made thing, an artifact of human creativity and culture. The church confesses that it is more than that—but it is not less. What that "more" comprises is a matter of faith, which is why it is an object of confession and not of empirical proof, much less logical demonstration. The phenomenal features of the texts of Old and New Testament are identical in kind to other members of the class of texts; in this respect, though not in others, the Bible is "like any other book".[2] There is no canonical text, no pericope or verse, not written by some or other human being. Not even the Decalogue as we have it is scribbled by the finger of God. The canon is a made thing; as such, it is also a human thing.

Be that as it may, the church has long held that, though the canon's human, historical, and cultural properties are real and relevant to understanding and interpretating it, they are not its defining feature. In perceiving them you have seen something true of the canon,

but you have not thereby understood what it most fundamentally is. For what it is—its ontology or nature—is a function of the divine will and action. Describing the canon apart from God is akin to describing a human embryo apart from her mother. Possible, but sorely incomplete and finally unilluminating. What the canon is, therefore, it is in relation to God. God is its preeminent maker and author, its first cause and special source, the prevenient artificer who has so fashioned these and only these texts that they are fit to serve as the embassy of his rule, the temple from which his voice resounds.[3] In accordance with his will, they thus constitute the one canon of Holy Scripture in and for the church, the one word of the one Lord for his one people.[4]

Neither Israel nor church fashions a theory of how this might be the case before it begins to read the prophets and apostles as God's word to his people. Rather, their writings appear in the midst of the people's life; the people eventually receive and transmit them in practices of public worship and private devotion; their presence and authority are taken for granted; and only then are theological questions about their nature and ultimate origins posed. After many centuries of liturgical use, personal piety, homiletical and commentarial interpretation, and subsequent theological reflection, the church developed a grammar by which to describe the scriptures' character as simultaneously divinely and humanly wrought. The primary term in that grammar is "inspiration". Following 2 Timothy 3:16, it was said that what set apart the canonical scriptures from all other texts, however true or edifying the latter might be, was that the former alone are inspired or breathed upon by God's Spirit. Unique among all texts, that is, they have God as their principal author or speaker. Whatever else is true of them, therefore, they successfully mediate the word of the Lord Christ to his body, the church; whatever other questions one might legitimately ask of them (the range of their competence; the nature of their historicity; the precision of their details; the relationship between them and sacred tradition), they are worthy of trust. From them the faithful are to learn; to them, in matters of faith and morals at least, they are to submit.

It is important to see, however, that while patristic and medieval thinkers presuppose the doctrine of inspiration, it is not load-bearing in their systems. It is not made to do much work. It functions more as a premise than as a conclusion, much less a detailed systematic theory.[5] It is not until the sixteenth century that inspiration assumes central importance; thereafter it is drafted into service as a major component of Christian doctrine (for obvious reasons).[6] Much of this service was fruitful: Protestant scholastic reflection on inspiration has a good deal to teach us, not least its rich integration with the doctrine of God and its sophisticated account of divine and human action.[7] But inspiration can be asked to do too much.[8] All too often there are only two major terms in bibliology: inspiration (God's role in Scripture's origins) and illumination (God's role in Scripture's reception).[9] The result is imbalance, superficiality, and misrepresentation. Imbalance, because inspiration ends up crowding out other aspects of God's work in making Scripture; superficiality, because the long and extraordinarily complex process of Scripture's coming-to-be is reduced by a single word to a simple once-for-all action; and misrepresentation, because the human-ecclesial component of Scripture is rendered either invisible or redundant. As I have already said, no account of the canon that elides or abjures its divine authorship can claim to understand it. But the reverse is true as well: to deny the *ecclesial* authorship of Scripture is equally to fail to grasp what it is. Under God, the church is the human author of the Bible.[10]

In search of a more adequate depiction of the nature and origins of Scripture, recent theologians have proposed new terms beyond the dyad of inspiration–illumination. These terms enlarge the theological imagination by offering a more detailed account of the discrete moments in the canon's production. The aim is for theological concepts to correspond to what we know to have been the case in the canon's actual history, human and ecclesial, and for these concepts to reinforce rather than undermine the church's confidence in the canon's determination by the divine will. Instead of inflating the reach of inspiration to mean *everything God did before the closure of the canon to make it his word*, inspiration is reserved for a

specific moment or activity, lest in coming to mean everything it ceases to mean anything.[11] In what follows I want to build on these recent suggestions, employing, refining, and expanding their proposed terminology. The goal is a revised theological lexicon for the artefaction of Holy Scripture.[12]

<p style="text-align:center">* * *</p>

Start from the end. When we speak of the millennia-long labor to fashion a canon for ecclesial use, what is the finished product to which we refer? We have already seen the answer: the texts of the prophets and apostles mediate that which they are not themselves in full: the living word of the living God. That is to say, they make (truly) present that which is not (fully) present, under the form and aspect of a sign, both as a fulfillment of the *past* promise of the Lord to his people and as a *present* promise to be fulfilled in full on the day of the Lord's return. God's word in Scripture is thus an eschatological reality: provision for the passage in the wilderness from initial deliverance to final completion in the Promised Land; manna for sustenance between the Red Sea and the River Jordan. Put differently, the canon is a function of the Great Commission because the canon belongs—exclusively—to the time between the times. In the absence of sin, there is no need for it; in the presence of Christ, it is redundant.[13]

The church has a name for such phenomena in its life: μυστήριον, *sacramentum*. As Robert Jenson (1999, p. 260) writes, however, "The liturgical or devotional reading of Scripture is not 'a sacrament' by any usual enumeration". Nevertheless, "the coincidence of heaven and earth, future and past, sign and *res* is the truth of Scripture's role and power: Scripture is indeed a 'mystery'". By "mystery" Jenson means a symbolic practice of the church, however great or small, that effects communion with the triune communion of persons, thereby offering a foretaste in this life of that which the baptized will partake of, without restriction or limit, in the life of the world to come. Whether or not Scripture is aptly denoted a sacrament, then, there is no question that it is *sacramental* in function. It is among the chief mysteries of the church's life of worship and mission.[14]

As such, Scripture is the sacrament of revelation. Its character, in turn, is more specific than its generic definition as a means of grace or a vehicle of divine presence. It bears to believers the grace *of the Word*; it is the medium, not of a mute deity, but of the *communicative* presence of the Son. When God is unveiled, he speaks; and between Ascension and Parousia, Scripture is that unveiling. To stand in the assembly when the sacred page is read aloud is to stand with Isaiah in dumbstruck contrition: "I saw the Lord sitting on a throne, high and lofty; and the hem of his robe filled the temple" (6:1). And just as Isaiah exclaims, "my eyes have seen the King, the Lord of hosts!" (6:5) so the faithful cry out, "my ears have heard the Christ, the Lord of heaven and earth!" The reading of Scripture is thus a promise that a day is coming when faith will be translated into sight: "For now we see in a mirror, dimly, but then we will see face to face" (1 Cor 13:12). Though the church cannot see her Lord, she does hear him. And his verbal bread is sufficient for the day.[15] For as St. Peter recognized, Christ alone has "the words of eternal life" (John 6:68).

A mystery of revelation: this is what the church, under God, produces in the canon. As a mystery, it is holy. What is holy is sanctified, set apart by the Lord for his sovereign purposes. Accordingly, John Webster (2003) argues that theology ought to deploy the term "sanctification" to the entire sweep of Scripture's existence. For "sanctification refers to the work of the Spirit of Christ through which creaturely realities are elected, shaped and preserved to undertake a role in the economy of salvation: creaturely realities are sanctified by divine use" (ibid., p. 26). But this work is limited neither to the beginning nor to the end of the canon's becoming, or even "to the text as finished product". Rather, "The Spirit's relation to the text broadens out into the Spirit's activity in the life of the people of God which forms the environment within which the text takes shape and serves the divine self-presence" (ibid., pp. 29–30). This broadening includes both "the complex histories of pre-literary and literary tradition, redaction and compilation" as well as canonization, interpretation, and illumination (ibid., p. 30). In short, the sanctification of Holy Scripture

by the Holy Spirit continues till kingdom come. It is a single operation across centuries, continents, and cultures to consecrate a book to be, and to be received in faith as, the word of God for the people of God. From ancient inciting events to stories handed down orally, from initial inscription to revision and redaction, from preservation and republication to ongoing liturgical use, from informal codification to memorization and translation, from synodal and episcopal lists to definitive canonization, from reception and approval to technologies of text production and private and personal use in the church: each and all of these moments, and many more besides, are superintended by God the Spirit in a temporally extended but unified action, working in and through the diffuse actions and wills of human beings to set apart the canon in and for the church.[16] As it is written, "no prophecy of scripture is a matter of one's own interpretation, because no prophecy ever came by human will, but [human beings] moved by the Holy Spirit spoke from God" (2 Pet 1:20–21). Sanctification is how this is true.

But not only sanctification. If that term denotes not only the production of Scripture but its ongoing use by—or better, its use by the Spirit in—the church, then what we need is a disaggregation of Scripture's sanctification into smaller moments, events, and activities. The simplest move is to draw a line, conceptually clear if historically tenuous, between the canon's creation and reception. Since it is the former that we are principally interested in, we will touch only briefly on the latter. The indefinite reception of the completed canon in the church's life may be understood as consisting in three elements: *transmissio*, *lectio*, and *illuminatio*. The church is tasked with (1) handing on the canonical scriptures, which includes taking care for their preservation, their translation, and their republication via various technologies. She is tasked further and above all with (2) *reading* the scriptures, privileging them in her life in all their irreducible textuality and linguistic particularity. But she is not to do so by her own lights, left to her own devices. Instead, the church is to read and to listen to Scripture (3) under the prayerful invocation of the Holy Spirit, by whose illuminating power alone the word heard in the words of the text will be "accepted . . . not as the word of men but as what it really is, the word of God" (1 Thess 2:13 RSV). Thus is the Spirit's work of sanctification accomplished in the church's reception of the completed canon.

It is this completion on which we are focused. Like Webster, Paul Griffiths (2011) has proposed that we appropriate a theological term native to another doctrinal locus and repurpose it here, in the doctrine of Scripture, in order to describe the creation or production of the canon. The term is "confection". In a commentary on the Song of Songs, Griffiths writes:

> This piece of scripture, like most, has a long and complicated history of compo-sition, redaction, edition, translation, commentary, and liturgical use, much of which is no longer accessible to us. "Confection" is a useful summary term for this process: to confect is to make something sweet and beautiful by judicious mixing of ingredients; it may also imply a co-making, an act of making in cooper-ation with other makers. The confectioner makes sweetmeats; the Catholic priest, it used to be said (the usage is archaic but elegant), cooperates with God and the people of God in the confection of the sacrament of the Mass; and the people of the Lord cooperate with the Lord in the confection of the canon of scripture as a whole and in each of its parts. (ibid., p. xxiii)

Griffiths goes on to elaborate a theory of Scripture's confection that includes not only various recensions of an initial document but continuous authoritative translations of whatever final form of the text—whether actual, hypothetical, or critically recomposed—is now taken to be a text's "original" version. For Griffiths, there are only versions; the "original" (almost always an imagined or critical construct) is the first of a series, and its location there is its sole precedence. Precedence, that is, with respect to the unqualified status of each and every version as truly Holy Scripture and therefore truly the word of the Lord to the church. For the Bible read aloud in the liturgy, whether in Latin or French,

Arabic or Swahili, is not prefaced or concluded by the proviso, "*a translation of* God's word". It is God's word as such.[17]

Griffiths is right about this, but we will leave his account of approved canonical versions to one side. It is his enlistment of "confection" for description of the divine-ecclesial activity of making the canon that I want to elaborate. The choice is felicitous in multiple ways. First, it draws on the language of the church's liturgy to describe another fixture of the same; for Holy Scripture is at once the cradle and the child of Christian worship. Theology does well not to forget this. Second, the language is sacramental, and as we have seen, the canon is a sacramental reality. Third, the sacramental analogue is the Eucharist, and the likeness between the visible word on the altar and the audible word on the page has long been a point of fruitful rumination for the church's teachers. Benedict XVI (2010, para. 56) writes in *Verbum Domini* that "Christ, truly present under the species of bread and wine, is analogously present in the word proclaimed in the liturgy". Likewise Henri de Lubac ([1959] 1998, p. 241) observes that "Scripture is bread, but for the Christian this bread does not become the living food that it ought to be until it has been consecrated by Jesus". Confection thus pairs well with sanctification, for as de Lubac observes, the scriptures are merely texts like any other, just as the bread and wine are food like any other, unless and until Jesus makes them otherwise.[18]

Fourth, then, "confection" recommends itself by its reference to a corporate human and ecclesial action in and through which the triune God enacts a work at once ordinary (a meal; a book) and miraculous (Christ's body and blood; the word of the Lord). It follows that no account of either the Eucharist or the canon will prove fitting that perceives divine and human activity to be mutually exclusive. In its production, Holy Scripture is both fully divine and fully human. There are no "gaps" in the record to fill in with exclusively divine action; nor are there parts of Scripture—its prehistory, its revisions, its editions, its translations, its collections—in which the work of the Holy Spirit is lacking. The canon comes from God; the canon comes from the church. In it we hear the voice of both. There is priority between the two: Ruth and Ezra, Acts and the Apocalypse are first of all the word of God; this is why they continue to be read in the church. But the priority of the divine does not erase the human. Ruth remains Ruth, and the vision of St. John was and is his and not another's. This reveals the disanalogy between the Supper and Scripture, at least on a catholic account of sacramental transformation. The authorial voice and human element of the Bible are not accidental to it. In this respect the sacramentality of the canon is akin more to consubstantiation than to transubstantiation.[19]

In any case, that is the rationale for employing "confection" to describe Scripture's creation, as a pair to its reception. But just as the latter calls for sub-division, so does the former. I will use two terms here, one familiar, one unfamiliar. The familiar is "inspiration". Inspiration includes the total sequence running from some original revelatory event or experience to its being written down in (more or less) fixed textual form. I suggest we see four discrete moments in this sequence, howsoever lengthy they may be in time: *revelatio*, *traditio*, *inscriptio*, *redactio*. For illustration, consider Ezekiel. "In the thirtieth year, in the fourth month, on the fifth day of the month, as I was among the exiles by the river Chebar, the heavens were opened, and I saw visions of God" (1:1). This is the revelatory moment. The moment of tradition is the handing-on, by word of mouth or formal teaching, of what was seen or heard, received or experienced. These are the folk stories of the patriarchs and judges, of Jesus and the apostles. In the case of an Ezekiel, one thinks of the circle of students or servants or fellow prophets who attend his message, spread it to others, or remember it after his death. Inscription is when that message is put into writing. Sometimes this might occur quickly; sometimes, not for decades or centuries. But at some point it occurs, whether or not during the lifetime of the subject or "author" in question. Finally, the text thus inscribed is revised: by colleagues (as with Ezekiel or Jeremiah); by delegates (as with St. Paul's letters); by a "school" (as with St. John's Gospel); by scribes of a more formal sort (as with the Law of Moses or the Psalms of David); by heirs who lay claim to a legacy (as with Isaiah).[20] This "moment" of redaction, best conceived as a complex

diachronic process, completes the whole series of events that began long before—say, by the river Chebar. The result: the Hebrew canonical book the church calls "Ezekiel".

Each of these moments or activities falls under the larger claim of inspiration. For inspiration is not limited either to the *writing* of a canonical text or to that text's *author's* mental or spiritual experiences or intentions. Inspiration is a predicate of canonical texts; inspiration therefore names the work of the Holy Spirit to produce each text just as it stands. If a text is something of a "pure" authorial product, lacking an inciting revelatory event, subsequent oral retelling, and serial revision—such as, plausibly, the letter to St. Philemon—then so be it: the old paradigm of the inspired author dictating a text in its finished form obtains in such a case. But this is the exception, not the rule. In nearly every case, the discrete texts that make up the canon contain, in their histories of production, most or all of the four aspects outlined above. And the doctrine of inspiration extends to each of them. God the Holy Spirit so moved in the lives, the minds, and the wills of each and every human person involved, however tangentially, in the creation of each and every canonical text—from Genesis and Exodus to the minor prophets to Acts and Romans to the catholic epistles—that the final product, the text as we have it, whether in the form of ancient manuscripts or an eclectic reconstruction, is in accordance with the divine will. The text, put simply, is what God wants it to be. It says what God wants it to say. In the formulation of St. Thomas Aquinas (1948): God's act, in sovereignly willing our free willing, is infallible but not coercive; he moves within us so that what we freely will to do is just what he wills that we do. Under conditions of sin, we may fail to do the good that God invariably desires; this failure he permits. But in the case of the canon, he permits no failure, no defection from his good intentions. The same logic that underpins the inscrutable δεῖ of the Passion—the claim that "it was necessary" for the Messiah to be handed over to the gentiles (Acts 17:3), to suffer and die and rise on the third day, for this occurred "according to the definite plan and foreknowledge of God" (2:23)—applies also to Scripture. Divine providence superintends its composition, from start to finish. This is what it means to say that it is inspired.

But inspiration is only one part of Scripture's confection: the familiar part. For inspiration concludes when the text in question is finished, and it is a mistake to extend the range of inspiration's meaning into the text's afterlife in the church. I suggest instead that we search for another term to denote this second part of Scripture's creation. Think of it along temporal lines: sanctification comprehends millennia, from Abraham to Christ's return.[21] The division between confection and reception lies somewhere in the middle patristic period (though given the informal character of canonization, it is conceivable to locate it much later, in the sixteenth century).[22] Inspiration then includes the whole history of Israel and the apostolic church, terminating traditionally with the death of the last apostle, though in my scheme continuing into the early second century (since certain canonical texts were likely written or definitively revised in that time).[23] But we need a means of describing the time, and in particular the activities, that occur between the apostles and, say, the Council of Chalcedon. There is overlap with what I have called the canon's reception, but in an important sense there is not a canon to be received during the second, third, and fourth centuries after Christ; the canon is a question and a task, not a fact. It is not merely given. This time is thus marked by *transmissio* and *lectio*; it is drawn forward by the Spirit toward canonization; but as a period of testing and discernment and debate, it includes also what I will call *probatio*. The term calls to mind the exhortation of 1 John 4:1: "Beloved, do not believe every spirit, but test the spirits to see whether they are of God; for many false prophets have gone out into the world".[24] By analogy, what the postapostolic church was called to do, distributed as she was across a vast range of locales, beset as she was by all manner of candidates for canonical status, was *probare spiritus si ex Deo sint*: to test the texts "to see whether they are of God". For, as we saw above, "no prophecy ever came by human will, but [human beings] moved by the Holy Spirit spoke from God" (2 Pet 1:21); yet as St. John warns, numerous pseudoprophets have gone out into the world. This means they must be tested, probed, discerned. The test for the spirits

(i.e., would-be Christian prophets claiming to speak by the Spirit) is plainly stated: "every spirit which confesses that Jesus Christ has come in the flesh is of God" (1 John 4:2). The test for the nascent patristic church used an expanded principle of discrimination: the Rule of Faith. What was ancient, of apostolic provenance, received by the many churches, and in accordance with the Rule was judged canonical; what was not, not.[25] This is the time of canonical testing, a kind of probationary period before the church was secure in her confession of those texts, and just those texts, that God intended to bear preeminence in the church's worship and teaching. By AD 500, the matter is largely settled—though questions continue to linger on the margins for some time.[26]

Probation terminates in a decision; this decision is canonization. I therefore propose, as a final term in our enlarged lexicon of Scripture's coming-to-be, that the overall process of transmission, probation, and canonization be named with help from another sacramental register: "chrismation". Chrismation is the completion of the sacrament of baptism; it is the sign and seal of the grace and holiness received in the waters that drown the flesh and raise one to new life in Christ. Anointed with oil, the baptizand is literally "christ-ed"—christened or chrismated—which is to say, effectually made a "little Christ" (Lewis 1952, p. 177). Baptism's fulfillment in sacramental anointing is the church's confirmation that the new believer who has just joined Jesus in the Jordan has also joined the Twelve in the Upper Room; there the risen Jesus breathes on them the promised Holy Spirit (John 20:19–23), pictured by St. Luke as fiery tongues descending upon the apostles at the feast of Pentecost (Acts 2:1–4). In this way the paired comings of Son and Spirit are symbolized in the twin, and thus inseparable, sacraments of initiation into the church.[27] In Orthodox practice, furthermore, the neophyte immediately receives the Eucharist for the first time, whatever her age, thereby completing the sacramental saturation of this newly adopted child of the Father by the persons and graces of Christ and his Spirit.

Following Webster and Griffiths' respective repurposing of sanctification and confection, I suggest we mix sacramental metaphors and use chrismation as a fitting image for the work of the Spirit to complete the creation of Scripture between its composition and canonization. Why? First, because inspiration ought to retain its proper scope and focus: the manifold processes leading to the production of the particular texts of the canon. Second, because reception, properly speaking, does not begin with the apostolic fathers; the issue of the canon and what it consists of is far more muddled than drawing a simple line between the apostles, on one side, and their immediate successors, on the other. Here, as elsewhere, the church's theological description of God's work in salvation history ought to correspond to what we reasonably believe actually to have occurred in that history. Consequently, third, a *theological* account of the canon's origins and formation calls for language that depicts the period between the final redaction of apostolic texts and the episcopal and synodal proposals of canonical lists as a domain of divine action. At the same time, fourth, this account must be ecclesial in character, for it is the church and the church alone, humanly speaking, that receives and debates and discerns and ratifies the texts that become the canon. Griffiths is therefore right to say that the church confects the canon, even as we, like he, make clear that this is God's work in and through the work of the church.

In sum, just as classical treatments of inspiration must be amended so as to incorporate activities beyond the singular writer dictating words (to an amanuensis) dictated to him (by the Spirit),[28] so treatments of inchoate scriptural collections and semiformal canonization should be expanded and deepened so as to offer a global depiction of this process in all its human complexity, historical detail, and ecclesial character *as* governed and determined by the Lord. The canon *becomes* the canon, and not just an undefined list of inspired books, by the work of God in the people of God.[29] This work is the chrismation of the scriptures. Before they were not a canon; now they are a canon: the holy medium of the Lord's word to the faithful. The canon finds its perfection or completion in this act of anointing, as it were: a descent of the Spirit upon the texts, sealing their contents and attesting their finality, a descent akin to the coming of the Spirit upon the elements in the Eucharist.[30] For this

reason the traditional quotation of Revelation 22:18–19 as applying to the canon as a whole is entirely appropriate: "I warn everyone who hears the words of the prophecy of this book: if anyone adds to them, God will add to that person the plagues described in this book; if anyone takes away from the words of the book of this prophecy, God will take away that person's share in the tree of life and in the holy city, which are described in this book". Read retrospectively, following the closure of the canon, this warning applies henceforth to all who might be tempted to alter it, whether by addition or subtraction.[31]

What Alexander Schmemann (1974, p. 75) writes of chrismation resonates wonderfully with its analogous deployment in bibliology: "Born again in the baptismal font, 'renewed after the image of Him Who created him,' restored to his 'ineffable beauty,' man is now ready to be 'set apart' for his new and high calling in Christ. Baptized into Christ, having put on Christ, he is ready to receive the Holy Spirit, the very Spirit of Christ, the very gifts of Christ the Anointed—the King, the Priest, and the Prophet—the triune content of all genuine Christian life". By extension, the *munus triplex* applies not only to baptized and confirmed believers but also to the inspired holy scriptures.[32]

* * *

The canon of Holy Scripture is a sacramental artifact of the church, produced by the undivided work of the Holy Trinity for the edification and nourishment of the church in her liturgy and doctrine. Its holiness is a divine gift, bestowed from above by Christ through the Spirit, on a par with the holiness of the Christian, received by grace through the sacraments of baptism, chrismation, and communion. As these sacraments are administered by men and women whom God uses as channels of his saving grace, so Scripture was fashioned by human hands through which God worked, invisibly, infallibly, and lavishly. However complex the process, the outcome is simple enough. And the church has always known the proper response: gratitude.

### 3. Speaking of Gifts

Gratitude is apt to a gift given and received. But before we label the canon a gift, we must know what we mean—or what others mean—by the term. In lieu of a survey or historical sequence, in what follows I will distill and summarize what I take to be the fundamental questions posed and concepts proposed by the philosophical and theological literature on the gift. The questions in particular will serve as branches along a theoretical decision tree. In this section I will mostly refrain from deciding which branches I take to be correct, though some such decision is unavoidable in the act of summary selection. The aim, though, is to wait until the third section to judge which choices are most fitting with respect to conceiving of Scripture as a gift.

The first question about the gift concerns what distinguishes it from a commodity, that is, an item bought and sold in the marketplace. At first glance the question might seem straightforward, but it turns out there is no strict or obvious demarcation between the two. As Marcel Mauss ([1925] 2016) observes in his foundational work, *Essai sur le don*, at least three obligations paradigmatically attend practices of the gift: the obligation to give, the obligation to receive, and the obligation to reciprocate. The implicit sense, commonly held at least in certain Western classes and communities for the last century or two, that gifts are by nature "free" in the sense that they are neither conditioned nor obligatory, is by no means common to other societies at other times. There is, accordingly, no prima facie reason to accord it normative status as a definition, much less as an ideal. The fact that this sense exists does, however, open up "the gift" as a discourse in which and with which to argue.

One argument that follows this disjunction is the nature of obligation in general.[33] Is the state of being obligated by another necessarily less preferable than not being so obligated? Does gift giving, at least most of the time, unavoidably obligate and (if obligation is less than ideal) thereby put others in a position that one would, or should, rather avoid?

Is obligation merely, or at least intrinsically, a form of or analogue to *indebtedness*? So that gift-giving is by its very form an act of power over another through which the givee is placed into a kind of peonage to the giver?—until, that is, such time as she "pays off" the debt accrued by (the obligation to receive) the gift.

Such are the worries of some theorists of the gift with respect to its presumptive obligatory powers. A broader way to state the concern is that the gift, on this traditional construal, is reducible to an economic contract written (however invisibly) into the unspoken norms and social expectations of a community. *Do ut des* (or *quid pro quo*) becomes the primary or sole form of relation between individuals as well as groups. How, from this perspective, can a gift ever be other than a burden? How, further, can it be genuinely gratuitous?

Griffiths (2009, p. 52) offers a strong contrast between two ideal types: "an economy of obligation and an economy of gift". Having described the former, he sets in relief against it an account of the latter. For "not all transfer [of goods] occurs within an economy of obligation", since some "occurs under the sign of gift":

> These transfers . . . are noncalculative and do not bring debt or any other obligation into being. Gift-givers in this economy do not concern themselves with the credit worthiness of those to whom they give; they do not consider whether those they gift are deserving; and they do not give with expectation of gratitude or any other return—though they hope that the gift given will be accepted as what it is, and as a result bring joy, and the flourishing that goes with joy, to its givees; they may also entreat their givees to accept what is given as it is given, just so long as they do not make the gift conditional upon such acceptance. Givers in this economy also often transfer inexhaustible goods, goods that when given and accepted are not lost to the giver. Those who accept a transfer within the economy of gift are, when they accept the gift as given, therefore obliged to nothing, and this makes the relation between giver and givee in the economy of gift simpler than that connecting those involved in the transfer of goods in the economy of obligation. In the latter, there is debt, discharge, and calculation as well as goods transferred; in the former, there are only giver, gift, and givee.[34] (ibid., p. 55)

Griffiths goes on to offer imperfect but nonetheless illustrative examples of noncalculative, non-zero-sum gift-giving: teaching; friendship; marriage; blood donation; almsgiving (especially alms given anonymously to strangers; ibid., pp. 61–62). Moreover, he specifies that what may appear to be conditions or "strings" attached to gifts are rightly interpreted as warnings, predictions, and entreaties, not as binding obligations. In this case, even the "return" of gratitude "is no more or less than acceptance and proper use, which is certainly a kind of reciprocity, but not one of debt or obligation" (ibid., p. 57).

Now consider an alternative account of gift, debt, and gratitude, from G. K. Chesterton (1924). After describing the shock of realization and joy at thankfulness for one's sheer existence, and the role this shock plays in the happiness and wisdom of saints and mystics alike, he writes:

> [T]he shortest statement of one aspect of this illumination is to say it is the discovery of an infinite debt. It may seem a paradox to say that a man may be transported with joy to discover that he is in debt. But this is only because in commercial cases the creditor does not generally share the transports of joy; especially when the debt is by hypothesis infinite and therefore unrecoverable. But here again the parallel of a natural love-story of the nobler sort disposes of the difficulty in a flash. There the infinite creditor does share the joy of the infinite debtor; for indeed they are both debtors and both creditors. In other words debt and dependence do become pleasures in the presence of unspoilt love; the word is used too loosely and luxuriously in popular simplifications like the present; but here the word is really the key. It is the key of all the problems of Franciscan morality which puzzle the merely modern mind; but above all it is the key of asceticism. It is the highest and holiest of the paradoxes that the man who really knows he cannot pay his debt will be for ever paying it. He will be for ever giving

back what he cannot give back, and cannot be expected to give back. He will be always throwing things away into a bottomless pit of unfathomable thanks. Men who think they are too modern to understand this are in fact too mean to understand it; we are most of us too mean to practise it. We are not generous enough to be ascetics; one might almost say not genial enough to be ascetics. A man must have magnanimity of surrender, of which he commonly only catches a glimpse in first love, like a glimpse of our lost Eden. But whether he sees it or not, the truth is in that riddle; that the whole world has, or is, only one good thing; and it is a bad debt. (ibid., p. 72)

Chesterton's point is threefold. First, what we have been given we could never repay. Second, this is the cause of praise and inexhaustible delight, rather than despair or resentment, not least because the giver of being is himself the infinite font of joy. And third—a distinctly Anselmian point—the infinite giver of a kind of infinite debt pays that debt, and then some, on our behalf as one of us; to such a double-giver no obligation could ever be bondage, for it is the highest form of freedom and flourishing.[35]

Turn, then, to the person of the giver broadly construed. What makes a giver a giver? What set of properties, dispositions, or obligations constitutes a giver as such—or, perhaps, sets apart the ideal or paradigmatic giver from approximations and failures to rise to that level? One strand of reflection on the gift, prominent in continental philosophy (and in theologians like Griffiths, as evidenced above), is that the giver must be free from compulsion: certainly of an external kind, possibly internal as well. Additionally, the giver must, on this view, be free of *interest* in more than one sense. The gift may not function as a sort of loan, collecting literal interest until the recipient "repays" with an equivalent gift "plus more" on top of the original "amount". In less strictly economic terms, the giver ought not to be *invested* in the use or issue of the gift. The negative example here is a parent "giving" (not loaning) an adult child a large sum of cash, then punishing the child—via disapproval, disappointment, resignation, or passive aggression—for not spending the cash wisely; that is, as the parent would have liked. In this way the ideal gift demands a wholly *disinterested* giver: both unbiased and removed from inserting herself into the designs and desires of the recipient.[36]

Whether or not the giver *conditions* the givee by giving a gift, may or should the giver's gifting be *conditioned* in some respect? Recent scholarship on St. Paul makes this question thematic for interpretation of his letters, specifically regarding the doctrine of grace. At the formal and historical level, John Barclay (2015, pp. 70–75) expounds six "perfections" of the gift, by which he means ways in which the concept and practice of the gift can be taken to their logical conclusion or even extreme. Each is a coherent account of the gift; though more than one may overlap, they need not entail some or all of the others; and no author, including Paul, should be presumed to mean any or all of the perfected concepts until exegesis shows it to be the case. Two of the six are especially relevant here: what Barclay calls the gift's "priority" and its "incongruity". The first names the total precedence of the giver's giving of the gift to the recipient's doing anything in relation to the giver so as to elicit the gift—including asking for it. "We love, because he first loved us" (1 John 4:19). The second perfection depicts the gift as incongruous, that is, as unrelated to and unmatched by the recipient's merit, or lack thereof. No relation obtains between desert and receipt. The gift is mine whether or not I asked for it, whether or not I deserve it, whether or not I am likely to use it well. In Barclay's words, such a gift may "be figured as one given without condition, that is, *without regard to the worth of the recipient*" (ibid., pp. 72–73, emphasis original). This is the strongest possible affirmative answer to the question of whether the giver, and thus the gift, ought to be unconditioned. This answer overlaps substantially with the traditional Augustinian and Reformed understanding of divine grace, though it is not coincident with it.[37]

One modest demurral from a Protestant theologian is worth noting, though. Edwin Van Driel (2021) affirms Barclay's proposal in the main but wants to amend one aspect of it regarding "conditionedness". For, as van Driel writes, "to say that God's election is not

based on any conditions in *us*"—for example, "lineage, genealogy, ethnicity, or biological connection" (Barclay 2015, p. 530)—"does not exclude that God's election may be based on conditions in *God*" (Van Driel 2021, p. 305, emphasis mine). He goes on:

> Theologically speaking, divine election can be characterized as a form of divine self-commitment. When God elects the other, God also elects Godself—that is, in election God commits Godself to be one's God, to be with and for the elect. If God elects not an individual but a people, God thereby commits Godself to the well-being of a lineage, of a genealogy—not because there is anything intrinsic to this lineage that conditions God, but because in electing a people God conditions Godself to be for this people and its offspring. (ibid.)

If we accept this argument as a plausible amendment to Barclay, then we may say the following: Even in situations of a gift's extreme incongruity, whereby nothing external to the giver conditions her giving, the giver herself may, by word or deed, condition herself, her future giving, and/or her ongoing relationship to the givee, just in virtue of giving a particular gift.[38]

There are, according to Barclay (2015, pp. 70–75), four other "perfections" of the gift. These are (1) *superabundance*, or the lavish nature of the gift; (2) *singularity*, or the complete characterization of the giver as generous without exception; (3) *efficacy*, or the infallible power of the gift to elicit precisely what it is given to do; and (4) *non-circularity*, or the postmodern "ideal" of a gift without return. Each of these raises alternatives for how to construe the gift, certainly with respect to God, grace, and Holy Scripture.

Consider efficacy. What does it mean, in whatever specified context, for a recipient to *fail* to receive a gift given? What concretely constitutes such failure? Are there situations in which it is impossible not to receive a gift? One common example here is creation. One cannot fail to be created, *having* been created, because there is no one "there" to resist the reception of being. At the same time, at least following upon a certain developmental threshold, failure to accept the gift of existence is possible: its name is suicide. Paired with creation is salvation. Is damnation a live possibility, or is God's will that all be saved (1 Tim 2:4) irresistible? If the latter were true, then in that sense the grace of Christ would be even less "optional" than the almost irresistible grace of creation. There is no suicide in heaven.[39]

This is efficacy in the strongest possible terms. Consider it in two other respects. First: In ordinary human situations, what is the ensemble of gestures, words, and actions—the choreography of reception and thanks—that together amount to *having successfully accepted a gift*? Doubtless it depends on context: time, culture, the type of gift, the relation between the persons involved. But such an ensemble always attends the giving and receiving of gifts, and there may, naturally, be unsuccessful attempts at performing the requisite moves, whether by misprision, ignorance, lack of practice, surprise, or flat disgust with or rejection of the gift. Second: Is part of the efficacy of the gift, as a function of its being received well, its issuing (whether with hope or with certitude) in certain behaviors on the part of the recipient? This might involve mere expression of gratitude; or action made newly possible by the gift; or a *change* in action on the part of the givee; or further circulation of the gift beyond the dyad of giver and givee. But whatever the intended behaviors may be, this aspect of efficacy is bound up with the earlier question regarding the interests of the giver. On one hand, it is difficult to imagine a truly uninterested giver. On the other, it is easy to see how the interests of the giver might easily, perhaps always, taint the gift with expectations that make of it something of a moral loan: *Do with this as I desire or command; if you do not, consequences will follow.* Those consequences might be mere disappointment, but they might also include penalties of a social, relational, legal, familial, or financial kind—most obviously a halt to future gifts!

This is the shadow side of the circulation of gifts. The positive side is that it is in the nature of gifts, across cultures, to create, foster, reinforce, or otherwise modify bonds within and among members of a community as well as between that community and other communities, including the community of heaven. Gifts, like sacrifices, are at once social and cultic, horizontal and vertical. They bind together that which always threatens to break

apart. Perhaps looking for a "pure" gift is a bit like looking for a "pure" exercise of coercion. Society cannot do without either, but neither is ever perfect in execution.[40]

Nevertheless, ideal types help to order and shape daily practice. Three further questions arise when we consider such types, each a concern within hailing distance of Mauss's threefold obligation and Barclay's six perfections, and all interrelated. Put simply: Is the gift given a *loss* to the giver? Is the gift itself a *scarce* good? And is the gift received *owned* by an individual or shared in common? Among contemporary theologians, Kathryn Tanner (2005a) has devoted considerable attention to these questions. As she writes, "what is notable about Christianity . . . is its attempt to institute a circulation of goods to be possessed by all in the same fullness of degree without diminution or loss, a distribution that in its prodigal promiscuity calls forth neither the pride of superior position nor rivalrous envy among its recipients" (ibid., p. 25). What such a vision generates, being rooted in an account of divine grace that is rooted in turn in a doctrine of the Trinity economically interpreted, is "a non-monetary, anti-monetary . . . market in goods" (ibid., p. 23). In other words, a non-competitive corporate life of abundant good and mutual fulfillment—without loss, without scarcity, without sacrifice, and without (the necessity of) reciprocity. Tanner elaborates:

> [G]iving to others should not mean the impoverishment of ourselves. Though we are not ourselves as an exclusive possession, though we are not only our own, neither are we dispossessed in giving to others—self-evacuated, given away. In human relations, as elsewhere in a theological economy, giving should not be at odds with one's continuing to have. Reciprocity of giving would certainly ensure this. In a human community where others are not holding their gifts simply for themselves, presumably what one gave away would come back to one from others. But reciprocity is not required to prevent self-sacrifice in giving here. Noncompetitive property and possession will do. What one gives remains one's property and possession and that is why giving does not come at one's own expense; one isn't giving by a giving away that might leave one bereft. . . . Rather than being in competition with our benefitting others, [therefore,] having becomes in this way the very condition of our giving to others. . . . As elsewhere in a theological economy, we are to give to others not out of our poverty but out of our own fullness. (ibid., 84)

Tanner's proposal, expanded in other works, is explicitly meant to be a sort of utopian intervention into the malaise and resignation of what she calls "the new spirit of capitalism" (Tanner 2019), that is, a market system dominated by finance. The ways in which both the economy of grace in Christ and the enactment of that grace in social spaces (including but not limited to the ecclesial) outside of market transactions are signs that there is a world beyond capitalism; which is to say, there are modes of having and giving and being and sharing that escape the reach of—if you will, *are unregulated by*—the market as we know it today. We are not (yet) at the end of history.

At any rate, Tanner's position usefully illustrates the difference made by how we understand the nature of the gift as a living event. The only remaining question concerns what she denies: reciprocity. Must there be a return? Must there be a countergift to the original gift? May gratitude alone satisfy the duty to respond to a gift in kind? Is St. Thomas Aquinas (1948) right that, though we owe the giver a fitting return, if we lack the means to "pay back" the gift, the *intention to do so* is adequate to the moral obligations of gratitude?[41] Or is Peter Leithart (2014) right to argue that, properly understood, Christian gratitude is due God alone, even and especially when fellow creatures bestow gifts upon us (whether or not we asked for them)? In such a case St. Paul is the exemplary "ingrate", for he exempts himself from the vicious circles of gift and countergift—debt and repayment, patron and bondage—by offering thanks *to* God *for* the gifts and persons of others.[42] After all, what good redounds to me from you has its ultimate source in God. The circle is not, however, hammered out into a straight line; in that case the altruists, or perhaps philosophers of the pure gift like Jacques Derrida (1992), would be right: no return; no gratitude; no conditions;

no interests; not even knowledge, either of the recipient or (ex hypothesi) of having given in the first place. On the contrary, Leithart interprets the New Testament to teach that the circle is not broken but rather expanded into an infinite circumference. So that, first, I know that in giving gifts to others, whether neighbors in need or even my enemies, I am through them giving gifts to Christ himself. And thus, second, for such good works or almsgiving, I neither expect nor demand a return in this life, whether of gratitude or of a countergift-plus-extra, certainly not from the poor, but also not even from my social or economic equal. This does *not* mean there will be no return, though. In this Jesus is one with Israel's sages: "He who is kind to the poor lends to the Lord, and he will repay him for his deed" (Prov 19:17); "But when you give alms, do not let your left hand know what your right hand is doing, so that your alms may be in secret; and your Father who sees in secret will reward you" (Matt 6:3–4). The return is intrinsic to the gift. The only question is when it will come and by whom.[43]

Which brings us back, once again, to thanks. Gratitude suffuses human life just as it does the life of the church. What form should such gratitude take in each of the contexts wherein we find ourselves? Is the gift incomplete without gratitude? Does gratitude *add* something to its recipient (that is, to the original giver)? Does it increase her honor or esteem or power or glory? Or is it merely a public or social recognition of a fact that is and remains true and complete regardless of such recognition? Should our answers to these questions change when we ask them of God? What, finally, is the meaning of the church's cry of thanks to God for the reading of the word in the liturgy? Does it denote the Bible as a gift? If a gift, what sort of gift? And what does this require of God's people, its collective recipient across time and space?

Having now laid the conceptual groundwork, these are the questions the next section aims to answer.

**4. The Gift of Scripture**

The premise or presenting issue for this article is the fact that the church gives thanks to God in response to the public reading of Scripture in the liturgy. This suggests the canon bears a gift-like status vis à vis the church. There is another way of coming at the question, though. The word of God is not owed humans, even apart from the fall. The presence of God's word to us is therefore and as such a grace, doubly so when that word imparts salvation to sinners. Nor need that word have been written down: there was when it was not, and nothing about itself or God's people requires its inscription. This makes of the Bible a kind of triple grace. Beyond being unmerited and salutary, Holy Scripture is lasting, reliable, imprinted not only on our hearts and minds but on parchment and paper. We receive it but also pass it on to each new generation.

Such continual passing-on is a hint, from another angle, of the gift-character of the canon. On one hand, we receive it not from God alone but from God through the church. On the other, we share somehow in the giving: whether merely by being the church in between our forebears and successors or more literally by fashioning, purchasing, sponsoring, annotating, donating, or bequeathing Bibles to and for the young—including our own children, grandchildren, and godchildren. In this sense the gift of Scripture is neither simple nor wholly asymmetrical. The *giving* of Scripture is grammatically and thus theologically participial: far from an ancient or a finished event, it is a living and unending process until the Lord's return.[44]

This understanding of Scripture's circulation among and between the many members of Christ's body across the generations accords well with our presentation of Scripture's artefaction in the first section. The canon comes from God, but at every stage and in every way its sanctification as God's word written is mediated by God's covenant people. Recall the language Griffiths (2011) proposed for the making of Scripture: confection. Here is Griffiths again, in his commentary on the Song, expanding on what his proposal means in practice for the church's life with the text. He begins by responding to the question, *What is a translated version of Scripture a version "of"?* Answer:

Among other things, of what the Lord says to his people. If scripture is the Lord's most explicit and complete verbal address to his people, then there is something that the Lord says to his people by way of scripture as a whole, and also by way of each of its proper parts, among which the Song is one. There is *a complex verbal caress with which the Lord delights and instructs his people, a kiss that he places upon his people's lips*—tropes especially apposite to the Song. This particular caressing kiss can be given to us here below only by way of words in some natural language or other, and since the depth and passion of the kiss is unfathomable, no set of such words can exhaust it. The words in which each version consists are successive attempts on the part of the people at various times and in various places to respond to *the Lord's verbal kiss*: this is true of the anonymous poets and scribes who put together the various successive versions of the Hebrew text; it is true of those who translated particular versions of that text into Greek and Latin and then subsequently into the languages of our times and places; and it is true too of those who have commented upon, preached about, or otherwise elucidated the words of the Song in any language. The confection of a scriptural book does not, therefore, end with the establishment of versions; those are its first yield, and they are inevitably and properly supplemented by commentary, which is confection's second yield, solicited by the first. Versions and commentary together are *the people's return of the Lord's kiss* (no kiss is given if one offered is not returned, as anyone knows who has kissed an unyielding pair of lips), and *the exchange of verbal kisses* will have no end here below. Versions and commentaries will, therefore, be endless. This book is one among those returned kisses. (ibid., p. xxvii)[45]

Scripture is a gift the way a kiss is a gift; and just as a gift's giving and receiving are a joint event, a kiss is not a kiss that is not mutual or shared. The church's repertoire of acts surrounding the scriptures—verbal, written, commentarial, homiletical, devotional, sacramental, liturgical—constitute the corporate response of the bride to the loving initiative of the bridegroom. We return Christ's kisses with our own.[46]

This imagery (a set of tropes, to be sure, but more than "mere" metaphors)[47] provides a useful point of contact with our exploration of the literature on the gift in the second section above, and thus may serve as a point of departure for considering some of the theoretical matters raised by a construal of the canon as a gift. So far my approach has been circumspect and somewhat dispassionate, wanting the terms and concepts to be clearly laid out before us. Having done that, however, I will now set forth a constructive and not only descriptively plausible rendering of the gift-character of the Bible. What follows, in a word, is a theological depiction of Holy Scripture as at once a divine and ecclesial provision to God's pilgrim people of the grace of God's word written.

\* \* \*

The word of God is an unmerited and gratuitous gift bestowed by the God of Abraham to his covenant children. It is subject neither to purchase nor to possession; it can be neither bought nor sold.[48] It is a gift of love imparted, first, to Israel according to the flesh; second, to gentiles baptized into Israel's Messiah; and third, to the world on whose behalf Israel was called and for whose sake the Messiah died and rose again.[49] It is a gift of mercy for sinners in need, part of whose desperate condition is not only ignorance of God's word but ignorance of their need for it, and thus an incapacity to seek or even to beg for that which alone might cure them.[50] This poor state, however, this incurable malady is not just sinsickness but wickedness. The loving Lord provides the remedy out of his bountiful goodness, but nothing from without compels him to do so. His act is not conditioned in that sense. It *is* conditioned in the sense that it is responsive *to* a condition, one that is contingent and not caused by himself. The medicine of immortality is an aid to mortals, not angels. Humanity would not need God's word in the form in which we have been given it

unless we were wretched. Nor, in a similar sense, need the shape of the canon be precisely what it is: whereas the Law and the prophets are, subsequent to Abraham's election and Israel's exodus from bondage, necessary in a manner of speaking, the apostolic writings are contingent in still a further sense. They *become* necessary following upon the Lord's good pleasure not to return in the lifetime of the apostles and their delegates. Had he come then, as presumably most of them expected, there would be no New Testament, and thus no "Old" Testament: only the scriptures of Moses and David, Solomon and Isaiah et al. The Lord's grace in extending the mission of the messianic covenant to include so many different peoples from so many different times and places is the Spirit-ordained "condition" for the gift of the canon as we have it.

Does the canon obligate? It does.[51] It obligates Jews through the laws and ordinances of the Torah and it obligates the baptized through proclamation of Christ's cross, what St. Paul calls ὁ νόμος τοῦ Χριστοῦ (messianic torah for eschatological gentiles: cf. Gal 6:2; 1 Cor 9:21).[52] Such obligations are no burden, however. As Jesus says: "Come to me, all you that are weary and are carrying heavy burdens, and I will give you rest. Take my yoke upon you, and learn from me; for I am gentle and humble in heart, and you will find rest for your souls. For my yoke is easy, and my burden is light". (Matt 11:28–30). Hence, the obligations of the canon—which is to say, the obligations of the covenant—are but the form of life proper to the covenant people. The joy of this life, further, is positively rather than negatively correlated with obedience to the wise commands of God, which (again, in Pauline language) is nothing other than the obedience of faith (ἡ ὑπακοή πίστεως; Rom 1:5; 16:26): "I will delight in your statutes; I will not forget your word" (Ps 119:16). The law worthy of the Psalmist's love (vv. 47–48, 97, 113, 119, 127, 159, 163–167), which is at once "the royal law" and "the law of liberty" according to St. James (2:8, 12), is summed up in the Messiah who imparts to his followers the new command that they should love one another as he loved them (John 13:34). In him they have eternal life (3:14–16, 36), not only as servants but as friends; for they are his friends if they obey his commands (15:12–15). Life and freedom, in other words, are found in obedience to the word of God. Which is another way of saying that they are found in obligation, which is not contrary to but a form of the gift. As Barth ([1942] 2010b) says, the command of Israel's God is grace in the form of an imperative.[53]

To live under the obliging grace of Christ is not, furthermore, just another form of indebtedness—unless, in Anselmian fashion, we are inclined to reshape and redeploy the conceptual field of "debt" as a sphere conquered and saturated by divine love, overflowing with liberality, universal remittance, and every debt paid with infinite interest. If, on this view, we "owe" God, then what is owed has always already been "paid" by Christ, to whom we are united in the sacramental life of the church. The actions of the head are the actions of the body. The "payment", if such there be, is an eternal circle of speech and reply, procession and praise, gratitude and joy in hypostatic and thus ecstatic form.[54] The only resulting "debt" for the members of Christ is the debt of love: for "love is the fulfilling of the law" (Rom 13:8).[55]

If the canon obligates, then, though it does not indebt, its giver is *dis*interested (which is to say, just)[56] but not *un*interested. The Lord gives generously but not to no end.[57] The purposes of Scripture are therefore bound up with this particular gift's efficacy. The divine fecundity of the word of God finds no better description than in the book of Isaiah:

> For as the rain and the snow come down from heaven,
>
> and do not return there until they have watered the earth,
>
> making it bring forth and sprout,
>
> giving seed to the sower and bread to the eater,
>
> so shall my word be that goes out from my mouth;
>
> it shall not return to me empty,
>
> but it shall accomplish that which I purpose,

and succeed in the thing for which I sent it. (55:10–11)

Since Holy Scripture is, by the church's testimony, the word of the Lord, we are right to conclude that its efficacy is potent, even infallible. This is one of its chief perfections, together with its priority, superabundance, and incongruity. Among other things, what it effects by the Spirit's power is the unveiling, advancement, and announcement of the advent of God's reign in the cosmos. It does not fail to do this; it never has and it never will. Its enactment looks like the ministry of Jesus. It looks, too, like the church, which is his body: it looks like gathering, worshiping, feasting, and serving. It looks like God's word visible, consecrated on the altar; it looks like God's word audible, read aloud from the lectern. When it slays sinners in the Spirit and raises them to new life in Christ, its effectiveness is on display. When it cuts to the heart and elicits faith and baptism, it is the irresistible gift of God's word at work in the world.[58]

Barclay's (2015) other two perfections are more doubtful as applied to Scripture. We have already seen that the "gift" of Scripture is in fact a constant missionary and generational *giving*, which suggests an essential circularity, against the terminal asymmetry of an anonymous and even unknowing postmodern giver. As for the "singularity" of Scripture, or rather of God, we may leave the question open. There is indeed a sense in which God, in himself as well as in the economy and therefore in Scripture, is unstintingly good and incomparably generous. But as Barclay points out, such a description at times lends itself to an account of God as immobile or lifeless, less than agential, "automatic" or even machine-like rather than personal.[59] Moreover, while the attribute finds support in Scripture (the Lord *is* good and *does* good: Ps 119:68), it also finds resistance in canonical teaching regarding the Lord's judgment, wrath, and punishment.[60] When the word of God mortifies, the Bible is a rod in his right hand; when the gospel crucifies, the scriptures are the vehicle of the divine verdict. We are right to interpret such actions as the *grace* of judgment and thus *not* a denial of God's and therefore of the canon's singular goodness. But such actions plausibly fall outside the scope of a strictly Platonic or generic deity whose beneficence is so "pure" as to be unstained by retribution or bloodshed.

The question of divine judgment raises in turn the question of Scripture's reception; in particular, whether the gift of the canon may *fail* to be received and in that respect fail to be given. We have already specified that Scripture will not fail to have a hearing, will not fail to hail the coming king in the presence of his glad subjects. But that does not mean the convocation will be universal, at least short of his epiphany. There are those who have not heard across a lifetime; there have been and will continue to be those who hear and do not respond with faith, obedience, or gratitude. This is just another way of saying that *living as Christ's body* is the proper mode of reception—a reception presumptively obligatory for all who hear, perhaps for every human soul after Pentecost—but that such reception, namely of God's word in Scripture, will always be wanting in some persons and communities.[61]

Practically speaking, we have already seen what constitutes the mutuality of gift and receipt with respect to the canon: the liturgical exchange of kisses between Christ and his bride. The "holy kiss" to which St. Paul exhorts the saints is here a figure of the reading and hearing of Holy Scripture in the church (cf. Rom 16:16; 1 Cor 16:20; 2 Cor 13:12; 1 Thess 5:26). Even when our verbal kisses amount to little more than the kiss of Judas, the kiss of Christ in the sacred meal is one and the same as that with which Christ fed the traitor at table: a gift we are free to receive, if only we will have it, and with it, himself.[62]

As we saw, though, the other side of this reception includes the church. In receiving the word of the Lord according to the Apocalypse I am entrusting myself not only to the words of the Seer as an instrument of the Lord's living speech or even to that word of which the Seer's words are an instrument. I am entrusting myself to the catholic and apostolic church as the corporate and diachronic recipient of the Revelation of St. John and, what is more, as its surety. In the terminology of the first section, the church is, in a more than nominal sense, the author of Scripture. The church's founders composed it; her predecessors preserved it; her councils canonized it; her leaders authorize its use in the present. To stand in a relation of grateful reception to God for the gift of Scripture, therefore,

is also to stand in a relation of grateful reception to God's people. It is to recognize the canon for what it is: an item of tradition, underwritten by the Holy Spirit. Just as I receive Christ from the celebrant, a member of the body standing in for the head, so I receive Christ's word from the church. For it is the church that has faithfully handed it on down through the centuries, right up to the point of my own hearing and faith and baptism and delectation in its delights—whereupon I join in the countless hands ("myriads of myriads and thousands of thousands": Rev 5:11) of the body of Christ and pass it on to those who yearn to hear it, both the living and the unborn.

Gifts create, maintain, renew, and strengthen social bonds. The gift of God's word builds the walls of the new Jerusalem. It effects the communion of the communion of saints: those at home with Christ and those still sojourning on earth, together with "every creature in heaven and on earth and under the earth and in the sea, and all that is in them" (Rev 5:13). With one voice they offer the only "return" possible for the gift of God's word: worship. When we read that "the four living creatures said, 'Amen!' and the elders fell down and worshiped" (v. 14), we see the one fitting but altogether necessary reply of the people of God to hearing his voice in their midst: *Amen*.[63]

The *Amen* of the body of Christ is nothing other than the voice of Christ's Spirit speaking within it, just as the sacrifice offered on the altar is nothing other than the self-offering of the Son to the Father in the Spirit. As St. Paul puts it, writing with St. Timothy:

> As surely as God is faithful, our word to you has not been "Yes and No". For the Son of God, Jesus Christ, whom we proclaimed among you . . . was not "Yes and No"; but in him it is always "Yes". For in him every one of God's promises is a "Yes". For this reason it is through him that we say the "Amen", to the glory of God. But it is God who establishes us with you in Christ and has anointed us, by putting his seal on us and giving us his Spirit in our hearts as a first installment. . . . [For] you are a letter of Christ, prepared by us, written not with ink but with the Spirit of the living God, not on tablets of stone but on tablets of human hearts. (2 Cor 1:18–22; 3:3)

The letter of the text is dead apart from the work of the Spirit. But by the Spirit's ministry—himself the preeminent *donum Dei* (see Acts 2:38; 10:45; John 14:16; cf. John 4:10; Eph 2:8; 2 Tim 1:6)[64]—the letter is alive with divine power. Through the word the Spirit writes Christ on the hearts of the faithful, so that all the promises that are Yes in him are Yes for them, too. When they hear and believe, when the Spirit implants the gift of the word in the soil of their hearts, their cry in reply is not them alone but Christ in them speaking back to God the Word that he is in the love of the Spirit, who unites them as one.[65] As Jesus prayed, those who believe on his name are one even as he and the Father are one (John 17:11, 22–23). So that, when believers give thanks to God for his holy word, they are merely reiterating that primal prayer to the Father by the Son, with whom they are one: "Father, I thank [εὐχαριστῶ] you that you have heard me" (11:41). Whereas the church thanks God that she has heard him, for according to Jesus's promise, his followers are "made clean by the word [τὸν λόγον] which I have spoken to you" (15:3). The circle of gifts is thus complete. Jesus bears witness to the Father that

> I have manifested your name to those whom you gave me [ἔδωκάς μοι] out of the world; yours they were, and you gave them to me [κἀμοὶ αὐτοὺς ἔδωκας], and they have kept your word [τὸν λόγον σου τετήρηκαν]. Now they know that everything that you have given me is from you [πάντα ὅσα δέδωκάς μοι παρὰ σοῦ εἰσιν]; for I have given them the words which you gave me [τὰ ῥήματα ἃ ἔδωκάς μοι δέδωκα αὐτοῖς], and they have received them [αὐτοὶ ἔλαβον] and know in truth that I came from you. (17:6–8 RSV)

By way of gloss: Those who belong to the Son (cf. 1 Cor 15:23; Gal 5:24) are gifts to him from the Father; they know all that the Son has is a gift to him from the Father, for the Son is himself from the Father (cf. John 5:26). The word of the Father, accordingly, is the word of the Son, and that word he has given in full to them that are his (cf. 2 Tim 2:19). They

have "received" it (ἔλαβον). What remains for them to hear, the Spirit will speak in the fullness of time, the same Spirit who will declare to them what is the Son's, for what is the Son's is the Father's (John 16:12–15). Or in the Pauline comment on the same reality: "all things are yours", for "you belong to Christ; and Christ belongs to God" (1 Cor 3:21, 23).

In short, the whole economy of grace is a storehouse of divine gifts. It is the domain of the supreme and unrivaled Giver, the Father, the Son, and the Holy Spirit. It is he and he alone, the Lord, who is the ultimate gift. The church receives him in many ways. As it is written, "When he ascended on high . . . he gave gifts to his people" (Eph 4:8). One such gift is the canon.[66] The artifact of Scripture has its place, therefore, within the triune economy. God's people have always been right to give thanks for it. In doing so, they thank the Lord for his grace: for his paternal care, his unrestricted solidarity, his accompanying presence. They thank him for speaking: for his wooing, his promising, his vowing—his kisses made of words. Most of all, they thank him for himself, since what the temple of Scripture bears to us just is the Lord; and from it his great and merciful voice resounds:

The Lord is in his holy temple;

let all the earth keep silence before him. (Hab 2:20)

**Funding:** This research is funded by the John Templeton Foundation, grant number G21000019.

**Data Availability Statement:** Not applicable.

**Acknowledgments:** My thanks to Justin Hawkins and Ross McCullough for their comments on a previous draft of this article.

**Conflicts of Interest:** The author declares no conflict of interest.

## Notes

1    Unless otherwise stated, all biblical quotations are from the New Revised Standard Version.

2    The infamous phrase comes from Jowett (1860).

3    I owe this image to St. Augustine of Hippo (1996, Pr.6).

4    "I am the good shepherd; I know my own and my own know me, as the Father knows me and I know the Father; and I lay down my life for the sheep. And I have other sheep, that are not of this fold; I must bring them also, and they will heed my voice. So there shall be one flock, one shepherd. . . . My sheep hear my voice, and I know them, and they follow me; and I give them eternal life, and they shall never perish, and no one shall snatch them out of my hand. My Father, who has given them to me, is greater than all, and no one is able to snatch them out of the Father's hand. I and the Father are one" (John 10:14–16, 27–30 RSV); "There is one body and one Spirit, just as you were called to the one hope that belongs to your call, one Lord, one faith, one baptism, one God and Father of us all, who is above all and through all and in all. But grace was given to each of us according to the measure of Christ's gift" (Eph 4:4–7 RSV). What this means for the synagogue is beyond the scope of this article; for now see Marshall (2013); Jenson (2019); Kinzer (2015).

5    This is evident in, e.g., Origen (2017) and St. John of Damascus (2022).

6    See, e.g., Preus (1957).

7    See, e.g., Muller (1993).

8    As can the concept of "canon" itself: see Abraham (1998).

9    Not that there are no other terms proposed. Most of them, though, are analogies applied to Scripture as a finished product, rather than to elements or moments in its creation or reception.

10   For fuller arguments see East (2021, 2022a).

11   As it arguably does in Jenson (2019, pp. 187–219) and Farkasfalvy (2018).

12   Analogies abound in what follows, to various sacraments and to the incarnation itself. I take for granted that analogies by definition encompass disanalogies, for the likeness of the comparison is imperfect. I only rarely pause to signal when this is the case, assuming it throughout. The utility of the analogies lies in their fittingness and in their capacity to encapsulate, in a familiar term, some action or process for which we lack a name. I leave it to readers to judge whether the proposed analogous terms fit the bill.

13   As Griffiths (2016, p. 57) writes, "scripture belongs to the devastation only; it had no place in paradise and will have none in heaven".

14   For substantive patristic reflection, see Boersma (2017).

15   "Verbal bread" comes from Leithart (2009, p. 207).

16  See further Gordon (2019).

17  I expand on Griffiths' argument in East (2021). Both there and here, Griffiths is a constant interlocutor and stimulant to theological reflection on Scripture's origins and exegesis as well as the concept of the gift.

18  See Rogers (2021) for further reflection on the semiotic valences of that "otherwise".

19  Is the eucharistic analogy even weaker than this? Is the canon more like the waters of baptism or the oil of chrism than the elements of communion? Perhaps. But it seems to me that, while oil or water may be blessed and thus made holy, the texts of Scripture are more than blessed elements that, in their *use*, effect what they signify. Something in the nature of the texts qua texts and their collection qua collection is ontologically and permanently sacramental from the inspiration and closure of the canon forward. Consider, by comparison, the proposal of Barth ([1938] 2010a), according to which the words of the canon *become* the word of God on concrete occasions of their reading. This move does, I think, make the Bible on a par with the water and oil of baptism and chrismation.

20  These examples encompass popular scholarly theories of canonical texts' histories of production without my committing to any of them being true. The point is that a proper account of inspiration is capable of incorporating any number of proposals about textual origins; it is not and never has been limited to the common image of the lonely author taking down the Spirit's words in the role of a secretary. For a sophisticated treatment of contested canonical authorship, see Johnson (2020).

21  The inciting event can be placed as early as Adam, if one prefers.

22  See, e.g., McDonald (2007); Metzger (1997); Barton (1998).

23  As with authorship, so with dating: this theory clarifies that Christians are not committed to, e.g., apostolic canonical texts being written by AD 75 or 100. There is thus no need to shape the evidence to fit the theory; the doctrine of inspiration is not affected one way or another by arguments about authorship or dating. Having said that, biblical scholarship on these questions is often poorly argued or question-begging, so neither the theologian nor the church is bound to accept whatever the latest consensus is. For a wonderfully dispassionate and thorough-going reconsideration of the dating of the New Testament, see Bernier (2022).

24  Greek: Ἀγαπητοί, μὴ π αντὶ πνεύματι πιστεύετε, ἀλλὰ δοκιμάζετε τὰ πνεύματα εἰἐκ τοῦ θεοῦ ἐστιν, ὅτι πολλοὶ ψευδοπροφ ῆται ἐξεληλύθασιν εἰς τὸν κόσμον; Latin: *carissimi nolite omni spiritui credere sed probate spiritus si ex Deo sint quoniam multi pseudoprophetae exierunt in mundum.*

25  The disanalogy being that texts deemed non-canonical were not thereby judged to be on a par with "false prophets". Instead they typically belonged to that vast and ever-growing category of edifying but fallible writings available but not per se necessary to any one believer.

26  It is important to add that this process is principally liturgical, as is its result. Whether or not a text is canonical is answered by whether it has been, and should continue to be, read aloud in public worship.

27  See further Jenson (1997); Schmemann (1974); Ramsey (1946).

28  An image portrayed in texts and not only in paintings: consider Gerhard's (2006, 1.18.2) description of biblical authors as "moved, driven, led, impelled . . . and controlled" by the Spirit.

29  See further Kelsey (2009) and de Sales ([1886] 1989), both of whom mark the important conceptual distinction between a list of authoritative texts and an authoritative list of texts.

30  In this way "epiclesis" might be preferable to "chrismation" as an analogical term of art for the closure of the canon. For the former is an invocation of the church through her priest, acting in the person of Christ, for the Father to send the Spirit down upon the elements of bread and wine, that they might become the body and blood of Christ (and so be the *gifts* of God for the people of God). Inasmuch as "confection" is our chosen word for the "making" of the scriptures, why not conclude the action with the very term that captures the climax of the eucharistic rite? Much would be gained by doing so; I would not protest someone making the case. My main reservation is that the epiclesis is a *moment* in the extended ritual of the meal, not a discrete ritual as such; whereas chrismation, though it follows the sacrament of baptism, is a sacrament unto itself. For this reason it can be described as an action, rendered grammatically as a verb; less so, or more awkwardly, in the case of the epiclesis.

31  It is therefore an urgent ecumenical problem that the divided communions do not agree on the contents of the canon. Far too often evangelical treatments of the doctrine of Scripture elide the magnitude of this issue, as though (1) canonization was a straightforward process and/or (2) what the canon comprises is self-evident.

32  I elaborate on this briefly in East (2021, pp. 59–62).

33  See further Adams (1999, chp. 10).

34  The notion of being "obliged to nothing" by reception of the gift is, first, one I will criticize directly in the final section; second, seemingly evidently rejected by the plain sense of Scripture; and, third, curiously suggestive as a formulation. Instead of the gift—in theological terms, the gift of grace, of Christ, or of the canon—entailing that one is obliged *to God*, the phrase implies a worrying *nihil* standing behind or contained within the gift. As I argue below, the anxiety attending an obliging gift seems, at least in part, to trade on the idea that obligations must be a kind of debt, or at least necessarily burdensome. I see no reason, though, why that should be the case.

35  See below for mention of St. Anselm and of some of his interpreters.

36  See further Griffiths (2009); Derrida (1992); Caputo and Scanlon (1999).

37    For more details, see Barclay (2015, pp. 79–150; 2020, pp. 137–48).

38    I think this a reasonable extension of van Driel's point, though I would choose a different way to describe what it means for God in metaphysical terms. The danger is making God subject to time or to events in time, including events caused by God.

39    Given the pairing here of creation and salvation, this is a good place to mention the intersecting and diverging work of Marion ([1982] 2012, [1997] 2013, 2016); Milbank (1995, 1997, 1999, 2003, 2022); and Hart (2003, 2017, 2020, esp. chp. 2).

40    See again Mauss ([1925] 2016); Leithart (2014); Barclay (2015). See also Anderson (2013); Brown (2012, 2015); Hyde (2019); Visser (2008).

41    For his discussion of gratitude and ingratitude, see Questions 106–7 in *ST* II-II.

42    See the full treatment of Paul (and Jesus) as "ingrates" in Leithart (2014, chp. 3). The paradigmatic instance of St. Paul's resistance to coming under the thumb of a patron is his letter to the Philippians. We do not learn of the apparently quite generous gift the church sent him in prison until the closing verses of the letter (4:10–20), and nowhere does Paul thank *them* for the money; he offers thanks to God alone *for* their gift (starting in 1:3). Indeed, Paul makes it sound as though he did not need the money and, hence, that the principal benefit of the gift is to the Philippians, enriching them spiritually and producing fruit that glorifies God.

43    The natural answer is that God will make the return in the next life, and exclusively there. The answer is surely more complex than this, though. Consider Mark 10:28–31: "Peter began to say to him, 'Look, we have left everything and followed you.' Jesus said, 'Truly I tell you, there is no one who has left house or brothers or sisters or mother or father or children or fields for my sake and for the sake of the good news who will not receive a hundredfold now in this age—houses, brothers and sisters, mothers and children, and fields, with persecutions—and in the age to come eternal life. But many who are first will be last, and the last will be first.'" And with specific respect to alms, see Tobit 12:8–10: "Prayer with fidelity is good, and almsgiving with righteousness is better than wealth with injustice. It is better to give alms than to lay up gold, for almsgiving saves from death and purges away every sin. Those who give alms will enjoy a full life, but those who commit sin and do wrong are their own enemies". Compare the description of Cornelius, a gentile God-fearer, in Acts 10:2–4: "He was a devout man who feared God with all his household; he gave alms generously to the people and prayed constantly to God. One afternoon at about three o'clock he had a vision in which he clearly saw an angel of God coming in and saying to him, 'Cornelius.' He stared at him in terror and said, 'What is it, Lord?' He answered, 'Your prayers and your alms have ascended as a memorial before God.'"

44    As participial, it is also participatory: the church cooperates in God's gracious action to gift his people with his holy word. This cooperation across time is the act of tradition, or παράδοσις; see further Lossky (1974); Lash (1978); Ker (1978); Kenneson (1989); Behr (1999); Louth (2005); Lattier (2011). It should be clear by now that every feature of my proposal, whether about the grammar of Scripture's production and reception or about the concept of Scripture as a gift, depends upon and supports a non-competitive understanding of divine and human agency. See further Tanner ([1988] 2005b).

45    This quotation shows that I have slightly modified Griffiths' concept of confection for my own purposes. I want to draw a hard and fast line between the making and the receiving of the canon. But Griffiths is right that no such line exists. He is further right that, if translation is participation in the confection of the scriptures, then properly speaking the making of the canon is a process without end; it is coterminous with the life and mission of the church militant. Its end is her end, which is nothing but the second coming of Christ. Nevertheless, it seems reasonable to say that there *is* a moment in the church's history when the canon qua canon is extant; subsequent to this moment, further versions of the canon may and will be produced, but they are *additional yields* following what Griffiths calls the "first yield". In my scheme I assign these proliferating yields to the ongoing reception of the canon on the part of the church and, more broadly, to Scripture's sanctification by the Spirit across time.

46    See further St. Bernard of Clairvaux (1979, 2008).

47    See Jenson (1999, 39n.41).

48    Cf. Isaiah 55:1–2; Revelation 22:16–17.

49    See further Griffiths (2009, pp. 176–82); cf. Volf (2005; 2010, pp. 3–40).

50    By this I do not mean that nature, and the desire for God proper to nature, is snuffed out. I mean, on one hand, that our sense of what we need is a sort of wandering in the dark; and, on the other, that *when* we beg for what we truly need (the saving word of Israel's God), it is the prompting of grace within us that makes it possible. In this way begging is itself a sign of the gift's presence to us. St. Augustine of Hippo (1998) stands behind much of this. See also Johnson (2007); Hauerwas (2004, p. 241).

51    The sacraments are exemplary here. For the sacraments are themselves undertaken in response to the Lord's command, and these commands are found precisely in the canon. In other words, the dominical instructions regarding Eucharist and baptism take the grammatical imperative. Nor would any but the most anti-sacramental call the sacraments burdens! Yet they are covenant obligations imposed by the canon. This example not only confirms that divine commands may be free of burden (gospel may take the form of law); it also underwrites continuity between old and new covenants, *both* of which contain, in Lutheran language, "law" and "gospel" in distinctive ways. All of this contra Griffiths (2018, esp. pp. 57–78).

52    See further Rudolph (2016). I take the phrase "eschatological gentiles" from Fredriksen (2017).

53    The lengthy discussion, to which I already alluded in a previous endnote, begins on p. 509 in *CD* II/2.

54    See further East (2022b).

55      I am only representing, not endorsing, this view and similar ones that make "debt" either (1) a major term in accounts of obligation to God or (2) a synonym for it. I want to acknowledge that it has been done (and that it has roots in Scripture: Matt 6:12; Rom 13:6–10; etc.), but I am not myself employing it in my own account. If, on a minimal definition, an obligation denotes something we ought to do (or keep from doing), I see no need in principle to redescribe the obligation in social or economic terms as a debt. Regardless, see further St. Anselm of Canterbury (1998); Hart (1998); Marshall (2011). Cf. Søren Kierkegaard (2009, p. 172): "love is perhaps best described as an infinite debt: when a man is gripped by love, he feels that this is like being in infinite debt. Usually one says that the person who becomes loved comes into debt by being loved. Along the same line we say that children are in love's debt to their parents, because their parents have loved them first and the children's love is only a part-payment on the debt or a repayment. This is true, to be sure. Nevertheless, such talk is all too reminiscent of an actual bookkeeping relationship—a bill is submitted and it must be paid; love is shown to us, and it must be repaid with love. We shall not, then, speak about *one's coming into debt by receiving love*. No, it is the one who loves who is in debt; because he is aware of being gripped by love, he perceives this as being in infinite debt. Remarkable! To give a person one's love is, as has been said, certainly the highest a human being can give—and yet, precisely when he gives his love and precisely by giving it he comes into infinite debt. One can therefore say that this is the *essential characteristic of love: that the lover by giving infinitely comes into—infinite debt*. But this is the relationship of infinitude, and love is infinite."

56      "For there is no respect of persons with God": Romans 2:11 KJV (cf. 2 Chr 19:7; Prov 24:23; 28:21; Acts 10:34; Eph 6:9; Col 3:25; Jas 2:1; 1 Pet 1:17).

57      "I do not nullify the grace of God, for if righteousness comes through the law, then Christ died for nothing" (Gal 2:21). Barclay (2015, pp. 439–42) interprets Paul to mean that mortal sin is a live possibility for believers; that is, having received the gift, one may fail to respond appropriately to its obligations and thus fail to keep it. It is in this sense that Barclay argues that grace is unconditioned but not unconditional: God's incongruous grace may be lost.

58      Note well the level of generality at which these claims are made. I am not arguing that in every instance the reading or hearing of the scriptural texts is an occasion of irresistible grace. I qualify such an extreme implication in the discussion below regarding the possibility of failing to receive the word for what it is.

59      By contrast, the works of David Bentley Hart (2003, 2013, 2017, 2019, 2022a, 2022b) amount to a sustained defense of this sort of Christian Platonism, with significant moral, hermeneutical, and doctrinal implications.

60      I have discussed some of the exegetical, historical, and theological questions raised by such teaching in East (2022c).

61      Advocates of universalism will disagree, though not necessarily with implications here, since the canon need not have a causal relation to postmortem purgation and deification of those who die in sin and/or in ignorance of Christ. See Hart (2019); but cf. Griffiths (2020). See also McCullough (2022) for a defense of what he calls "indeterminist compatibilism", which might be usefully deployed in the doctrine of Scripture and its (sometimes failed) reception.

62      A whole theology of the gift, differentiated by confessional lines, could be built upon the reception history of the Gospels' depiction of the presence of Judas at the Last Supper. See, e.g., Calvin ([1845] 2008, 4.17.33–34).

63      For further liturgical reflection, see Schmemann (1973); Zizioulas (1985).

64      See St. Augustine of Hippo (1991).

65      There is rich fodder here both for a doctrine of the Spirit as the *vinculum amoris* and for a doctrine of the church as the *totus Christus*.

66      See Webster (2012, chp. 2).

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
