# Peer review of "Is Scripture a Gift? Reflections on the Divine-Ecclesial Provision of the Canon"

_religions, doi:10.3390/rel13100961_

Round 1

Reviewer 1 Report

1. Several sources on gift/grace and reciprocity could have helped buttress the author's arguments further: e.g., Seneca, De Beneficiis; and contemporary biblical scholars, all writing before Barclay's Paul and the Gift came out: such as James R. Harrison's Paul's Language of Grace in its Graeco-Roman Context (WUNT, 2003); Troels Engberg-Pederson's “Gift giving and Friendship," Harvard Theological Review 101 (2008); B. J. Oropeza's "The Expectation of Grace," Bulletin for Biblical Research 24 (2014); and David deSilva's Honor, Kinship, Patronage, and Purity,  recently updated to a second edition. The author might want to add one or more of these to footnotes and/or bibliography as is available to him/her. 

2. The author uses "givee" rather than recipient or another more standard term. The author perhaps should either use a more standard term or justify the use of "givee" in a footnote. 

Author Response

These are apt and well-received comments. I was trying to strike a balance between "lit review" and "serious engagement with major research of direct relevance to the theological topic at hand." I hope I struck the right balance, but these uncited sources are clearly germane to the larger conversation.

Reviewer 2 Report

Thank you for the opportunity to read your very interesting article, “Is Scripture a Gift? Reflections on the Divine-Ecclesial Provision of the Canon.” It is an important question that has significant implications for the church’s response to the reception of scripture. I hope that my revision suggestions below will help to make your article clearer and more persuasive.

My main concern with this article is that it poses a question without making clear, both at the beginning and throughout, who might disagree with your answer. You have a very interesting list of references, some of which have generated really contentious debates. But most of your engagement with your sources seems to build on their arguments without establishing the debate. I think there are two options here.

The first is to establish more clearly in the introduction how your argument is a response to a current discourse about scripture. Who would disagree? What other definitions of scripture are being debated? Is there a reason a more nuanced understanding of inspiration and confection/chrismation requires seeing scripture as a gift? One opportunity is to engage more deeply with Milbank’s theory of gift and participation, which can help to extend your question, for example, to the following: “Is scripture still a gift if we participate in its confection?” Currently, the relevance of section 2 is a bit unclear, but if you set up your argument as a response to Milbank’s critique of Reformed theology, it may help to make the relevance of your question clearer.

A second option, one that might require a more significant revision, is to engage more deeply with the theoretical and philosophical discourses about the gift. If the purpose of section 3 is “to offer a guide to the spectrum of theories on the gift and gratitude” (p. 2, lines 69-70), I’m not sure that an engagement primarily with Griffiths, Chesterton, Barclay, and Tanner is sufficient. It begins rightly with Mauss, but what of the more significant debates among Derrida, Marion, and Milbank? My sense is that the more you engage with these primary theorists of the gift, the more your central question might end up shifting away from whether scripture is a gift to whether the gift of scripture can obligate without debt. That seems to me where your article was headed in the final section of your argument (pages 16-17). Should you decide to reframe your question and article to address that specific point about a gift that obligates without indebtedness, I think your article would be much more relevant to the discourse surrounding gift theories.

Whichever option you choose, I would suggest beginning each section with a clearer section intro that states both the purpose of each section and how it contributes to the overall argument. The more you can connect the dots for your readers at the beginning of each section, the clearer and more persuasive your argument will be.

I look forward to reviewing a revised version of this paper.

Author Response

After consulting with academic editor of this SI, the first three reviewers said the paper was publishable in its initial form, with one of them saying it is a stellar piece of theology that deserves wide readership. After we revised the ms into final form, a fourth report rolled in. The reviewer was not as favorable, inviting substantive revision. Reviewer 4 seems to want a different type of paper, but nothing in his comments leads academic editor to believe the paper has significant shortcomings as is.

I appreciated the final reviewer's comments, which were substantive. We were in the final stages of the revision based on three reviewer reports that had already been submitted. Each of those reviewers was positive about the manuscript, stating that it was publishable in its initial form (one calling it a stellar piece of theological scholarship).
Please also see AE's comments below: "I concurred as editor, and therefore it seemed unfair to me to ask the author to make significant changes at that late stage in the process. Had the first three reviewers been more ambivalent, or had I myself had concerns about the quality of the paper, I would have asked the author to revise in response to the fourth reviewer's comments, but that wasn't the case. Nevertheless, I know the author was appreciative of the care with which reviewer 4 read and commented on the ms, and I suspect those comments will shape the author's work going forward. "

Reviewer 3 Report

This is fine paper that makes a contribution to the church’s discussion of the nature of the canon in our life as the church, and rightly the argument depends on its location there, the place where the canon functions as canon. As I understand it, the baptized (chrismed) word presents itself as a sort paraclete, presence of the Son in out sojourn, part of a Trinitarian dispensation.

The argument presents itself, first, as an account of how one should understand the role of Scripture, one in which it is graciously formed by the Spirit through human processes. As the argument progresses, the process of “reception,” gave me slight pause—transmission, reading, illumination, seemed rather singularly directed at the recipient, and “confection” appeared as simply a product of the past (cf. lines 344-45). Nevertheless, Herringer’s final section aptly draws in the creative engagement of the Spirit-formed community with the canon (H. notes this dynamic in fn. 45).

Throughout, though, I was drawn to ask (perhaps beyond the paper’s scope), what is the church’s proper disposition before the text. The focus on identifying the grace of Scripture in terms of incongruity and efficaciousness could lead one to imagine one’s passive role in receiving that “kiss” that H. references. Put another way, to what degree is the Spirit-filled church’s posture before the text submission, or critical engagement? Certainly, the Spirit is not at odds with the Spirit, but does the text function as the location of the church’s negotiation of the tradition, a location of divine dialogue, where certain aspects of the text can be questions and lamented, such as the old, but still rightly noted concerns of genocide in the text? Perhaps this is beyond the scope of the paper, but such an argument might contend with how, in the complexity of the text, the canon comes as the word of God. Put another way is the temple from which God’s voice resounds (line 107), like Solomon’s or Herod’s, built from wealth extracted from conquered peoples, taxes, or heavy labor, or is such a suggestion blasphemous. Maybe this is too heavy-handed, and likely it is unfair for what is really a wonderful paper, but the paper did draw me towards further consideration of the depth of the canon’s humanness and to ask how we understand a proper posture before it, coming as Spirit-formed interpreters.

I also found compelling H.’s discussion of “gift,” and I appreciate H’s use of Barclay’s perfections. H. rightly asserts the importance of the canon’s efficacy, and helpfully, H’s fn. 58, rightly denies that every instance of reading Scripture results in an irresistible grace. Yet, it is precisely the need to state this that points to something else that needs to be filled in. What are the conditions then of efficacy? Sociologically, and not theologically, speaking—and perhaps this is the wrong register—reading Zoroastrian texts, for instance, does not have efficaciousness for me because I am not formed into a community wherein those texts are authoritative discourse. This does not, of course, account for the work of the Spirit in the word’s going out, but I also would maintain that such an account requires a theological accounting for the formation of communities and situations guided by the Spirit so that the word is prepared to be received, read, and negotiated.

I did wonder about the discussion concerning the conditioned nature of grace. H. focused on incongruity in this regard. Certainly, incongruity concerns pre-conditions, but it seems that the weight of the modern argument about the “pure gift,” though not excluding incongruity, falls on the issue of circularity, expectation. Is there the conditioned expectation of response. This is what is behind Griffith example, noted by H., of the parent giving to children. Perhaps to be clearer there should be an additional category beyond non-circularity that reference the expectations of use, though not requiring repayment. One could imagine a parent giving to a child incongruously, not with expectation of being paid back, but with expectation that a gift be used well and in keeping with the virtue expected by the parent of the child. Perhaps this is understood as reciprocity or not, but one could certainly argue that Scripture is a gift that is not only efficacious, but that has an expectation, that expects to be received and acted upon. Nevertheless, the example of the parent is telling because, ideally, the basis of the relation is not the exchange of individuals, but of a social reality in which “selves” are linked, connected, and dependent within webs of mutuality and charity. Discussion of gift, in this way, requires a discussion of relationality and notions of the self.

More mundanely, note the typo on line 585 “whether I not I.”

These are simply the thoughts that came to mind as I delightedly read through this manuscript. I offer them hoping they may be useful to the author. As I have noted, the work offers a considerable contribution to our thoughts about the canon and its place within the community, giving an appropriately and well-developed theological account for the canon as artifact, sacrament, and gift for which the church gives God thanks.

Author Response

I have nothing except thanks and appreciation to say in reply. What these comments from the reviewer prompt in me is not so much revision as initial thoughts toward a sequel paper. So much to consider and reflect on. Many thanks.

Reviewer 4 Report

This is an excellent work of Christian theology - stellar, really. My only question is whether Religions is in the habit of publishing Christian theology. The paper simply assumes a Christian and theological audience. Is that beyond the scope of the journal? So long as it fits, this is exemplary work and should be published. But if there's any concern about whether the journal should be publishing something of this sort without at least a few words of contextualization, then the editors should be reflecting on that question.

Author Response

Many thanks. Yes, this article is an odd fit with the journal, but thankfully, the editors have been clear that they are happy to publish something frankly theological such as this. With happy precedent in scholars like Douglas Campbell!